# Postprandial Glucose Control in Type 1 Diabetes: Importance of the Gastric Emptying Rate

**DOI:** 10.3390/nu11071559

**Published:** 2019-07-10

**Authors:** Roberta Lupoli, Federica Pisano, Brunella Capaldo

**Affiliations:** 1Department of Molecular Medicine and Medical Biotechnology, Federico II University, 80131 Naples, Italy; 2Department of Clinical Medicine and Surgery, Federico II University, 80131 Naples, Italy

**Keywords:** postprandial glucose, gastric emptying, meal, premeal insulin

## Abstract

The achievement of optimal post-prandial (PP) glucose control in patients with type 1 diabetes (T1DM) remains a great challenge. This review summarizes the main factors contributing to PP glucose response and discusses the likely reasons why PP glucose control is rarely achieved in T1DM patients. The macronutrient composition of the meal, the rate of gastric emptying and premeal insulin administration are key factors affecting the PP glucose response in T1DM. Although the use of continuous insulin infusion systems has improved PP glucose control compared to conventional insulin therapy, there is still need for further ameliorations. T1DM patients frequently present a delayed gastric emptying (GE) that produces a lower but more prolonged PP hyperglycemia. In addition, delayed GE is associated with a longer time to reach the glycemic peak, with a consequent mismatch between PP glucose elevation and the timing of premeal insulin action. On this basis, including GE time and meal composition in the algorithms for insulin bolus calculation of the insulin delivery systems could be an important step forward for optimization of PP glucose control in T1DM.

## 1. Introduction

A large number of epidemiological and pathophysiological studies have demonstrated that postprandial (PP) hyperglycemia contributes greatly to overall glucose control assessed by HbA1c, and increases the risk of micro- and macrovascular complications in diabetic patients [1,2,3,4,5]. Whether PP hyperglycemia affects cardiovascular (CV) outcomes independently of overall glucose control is still a matter of debate [6]. A great number of clinical and pathophysiological studies have consistently demonstrated that rapid and brisk fluctuations in blood glucose exert harmful effects on the vascular system. [7]. Despite solid clinical and mechanistic evidence, intervention studies evaluating the effects of reducing PP hyperglycemia on CV outcomes in subjects with impaired glucose tolerance or type 2 diabetes have provided contradictory results [8,9,10]. To date, no long-term intervention study has been performed in T1DM patients. However, clinical studies in T1DM have shown that the endothelial dysfunction and the increased oxidative stress induced by a rapid increase in blood glucose can be reverted through glucose normalization, indirectly confirming the clinical importance of effectively controlling PP hyperglycemia [11]. As a matter of fact, the control of PP hyperglycemia is a recognized important therapeutic goal in the management of diabetes with a target 2-h PP glucose level < 140–180 mg/dL according to various International Guidelines.

In patients with T1DM, achieving an optimal PP glucose control remains challenging, even if the growing use of continuous glucose monitoring (CGM) systems provides a detailed picture of daily glucose fluctuations, including those occurring in postprandial periods. Indeed, PP hyperglycemia is the result of many interconnected factors, such as the quantity and quality of macronutrients [12], and the characteristics of insulin regimen [13]; all of which affect both the extent and timing of PP glycemic response. An emerging key factor is the gastric emptying (GE) rate [14,15,16,17,18,19,20,21]. According to the literature, approximately 40–50% of T1DM patients present altered GE, which is known to influence the PP glucose pattern. We have recently demonstrated that young T1DM patients with delayed GE present a different “shape” of post-meal glucose response characterized by a longer time to reach the glycemic peak compared to those with normal GE; this glucose pattern would require appropriate pre-meal insulin adjustments [20]. Individualizing pre-meal insulin administration taking into account the meal characteristics as well as the rate of GE could be an important step forward towards the optimization of PP glycemic control in patients with T1DM. The object of this review is to examine the main determinants of PP glucose control in patients with T1DM, with a specific focus on GE time and its impact on PP glycemia. The therapeutic implications of delayed GE in relation to the management of pre-meal insulin will also be discussed.

## 2. Gastric Emptying and Postprandial Glucose Control

### 2.1. Physiology of Gastric Emptying

GE is known to depend on the coordinated motor activity of the stomach and the upper small intestine, controlled by electrical slow waves generated by the interstitial cells of Cajal. The regulation of this system is complex and hinges on the inhibitory feedback arising from the interaction of nutrients with the small intestine, and the modulation by vagus nerve and gut hormones, namely GLP-1, cholecystokinin, peptide YY and ghrelin [22]. The GE rate is greatly influenced by the composition (solid or liquid), energy, and macronutrient content of the meal. Indeed, GE of solids is characterized by a biphasic pattern: a “lag phase” due to the solids being broken down into smaller particles, followed by a “linear emptying phase” in which GE time is approximately linear. In contrast, GE of liquids begins immediately and is directly proportional to the volume of the stomach content [23]. Also, the calorie content of the meal can influence GE: the increase in meal size and calorie content, with an equal macronutrient composition, significantly slows down GE [24]. As for the macronutrient composition of the meal, there is clear evidence that dietary fibers, fat, proteins and low GI foods tend to slow down GE [25,26,27] either through their influence on physical and chemical composition of the chyme or through the stimulation of gut hormones [28].

The rate of GE is modulated by acute changes in blood glucose concentrations, since hyperglycemia, even at physiological levels (~8 mmol/L) delays GE whereas hypoglycemia is associated with prompt acceleration of GE [29]. These changes have been interpreted as an additional mechanism of glucose regulation, i.e., the entry of glucose into the small intestine is slower in condition of hyperglycemia and accelerated in case of hypoglycemia to rapidly restore normoglycemia. It is also important to underline that the relationship between blood glucose and GE is bidirectional: ambient glycemia influences GE time and, reciprocally, GE rate impacts on PP glucose levels. In healthy subjects, GE is estimated to account for about 35% of the variance in glycemic response following oral glucose or a carbohydrate (CHO)-rich meal [30].

### 2.2. Gastric Emptying and Type 1 Diabetes

Diabetes is frequently associated with motility dysfunction of the upper gastrointestinal tract [31,32]. Indeed, T1DM patients can present with altered GE, which ranges from delayed GE—with a prevalence of 40–50% [12,24,25,26,27,28,29,30,31]—to, very occasionally, accelerated GE [33]. This variability is likely due to the heterogeneity of study populations with regard to diabetes duration, presence of chronic complications, and differences in the methods to assess GE. Moreover, the absence of upper gastro-intestinal symptoms can make the recognition of this condition particularly difficult, with consequent underestimation of its prevalence. Several mechanisms are held responsible for GE alterations in T1DM patients, including autonomic neuropathy, changes in gut hormone and neurotransmitter patterns, and fluctuations in blood glucose levels [31]. Autonomic neuropathy has long been recognized as a main cause of altered GE. Indeed, GE dysfunction is itself a manifestation of autonomic neuropathy and is frequently associated with other chronic complications of the disease. Indeed, in the DCCT/EDIC and in other cohorts of T1DM patients, 40–50% of patients showed delayed GE associated with severe retinopathy and cardiovascular autonomic dysfunction [19,34]. However, the relationship between delayed GE time and autonomic dysfunction has not been consistently proven [23,35,36,37,38,39].

We have recently demonstrated the presence of delayed GE in a relevant proportion (36%) of young T1DM patients without any clinical sign of chronic diabetic complications [20], indicating that an impairment of GE can occur quite early in the natural history of the disease–before clinical signs of complications in other districts become evident. Of note, these patients did not present any alterations in the level of gut peptide hormones, indicating that in T1DM patients the role, if any, of these hormones in the impairment of GE is likely to be marginal [20,40].

It has been widely demonstrated that GE disorders have a major impact on PP glucose concentrations in T1DM patients, in terms of both overall glucose response (area under the curve) [30,41] and magnitude of PP glucose excursions [15,26,33,42]. Using CGM, Parthasarathy et al. showed that a delayed GE was associated with higher glucose concentrations over the entire day [34]. Indeed, mean glucose values over the entire glucose monitoring time was directly associated with insulin dose and longer GE time, and inversely correlated with CHO consumption. In particular, each 10% increase in GE time resulted in a 0.7% increase in mean glucose value. PP glucose profile is particularly unstable in T1DM patients with gastroparesis, who are likely to present a condition of “gastric hypoglycemia” due to a mismatch between exogenous insulin action and the rise in blood glucose [43].

Not only gastroparesis but also mildly delayed GE can affect PP glycemia. In our study, T1DM patients with delayed GE time showed a different “shape” of post-meal glucose response that was characterized by a longer time to reach the glycemic peak compared to T1DM patients with normal GE (the average difference was 27 min). Interestingly, glucose peak concentration and PP glucose response did not differ between the two groups; this indicates that GE impacts the timing of post-meal glucose increments, as supported by the direct association between GE time and time-to-peak glucose [20].

Whether the use of prokinetic drugs has beneficial effects on PP glucose response in T1DM patients with delayed GE is not clear. Extant research in patients with gastroparesis demonstrated the potential benefits of prokinetic drugs in accelerating GE and in reducing hypoglycemia due to the mismatch between exogenous insulin and PP glucose levels [22]. No data is available on the efficacy of prokinetic drugs in T1DM patients with mild alterations of GE rate; in the meanwhile, in these patients there is no way to handle PP hyperglycemia other than to adopt the insulin regimen that best matches their PP glucose profile.

## 3. Other Factors Influencing Postprandial Glucose Control

### 3.1. Meal Characteristics

The macronutrient composition of the meal can influence PP glucose response either directly as in the case of CHO, or through the modulation of gastro-intestinal hormones and, hence, their impact on GE [12,16,44]. In healthy subjects, the increase in PP glucose depends mainly on the quantity but also on the quality of the ingested CHO; according to their ability to raise glucose levels, foods are classified as high glycemic index (GI) and low GI [45]. This has further evolved to the concept of glycemic load (GL), which considers both GI food and the amount of CHO contained in a given portion [46]. High GI foods are characterized by fast CHO absorption and a rapid increase in blood glucose, whereas low GI foods, because of their content in fibers, and/or protein, and/or fat, are digested and absorbed more slowly, with a more gradual increase in PP glucose [47,48]. In addition, in healthy subjects, both dietary protein and fat tend to lower PP glycemic peak and prolong glycemic excursions, mainly due to the ability of protein to stimulate insulin secretion and the ability of fat to slow down GE [44]. Over recent years, growing attention has been paid to the influence of food order on PP glucose levels; indeed, consistent evidence indicates that consuming protein or fat prior to a CHO-rich meal results in a lower glucose response compared to when all macronutrients are consumed together [49].

In patients with T1DM, the CHO content and overall composition of the meal impact on PP glucose levels and, consequently, on the timing and dose of premeal insulin [50,51]. Studies evaluating the effects of GI on PP glycemia in T1DM patients reported significant differences in blood glucose levels, with low GI foods producing lower glycemic response [52,53]. In a long-term (6 months) study, a low GI/ high fiber diet decreased blood glucose levels, HbA1c, and the rate of hypoglycemic events compared to a high GI/low fiber diet with similar macronutrient composition [54]. With regard to the determination of pre-meal insulin dose, several methods have been developed over the years. CHO counting has for long been a key dietary strategy for improving PP glycemic control; however, since the amount of CHO alone does not entirely account for PP glycemic response [55], novel insulin dosing algorithms based on more complex parameters have been proposed, such as the GI/GL of the meal [56], the fat and protein content, or the energy content of the portion consumed (Food Insulin Index) [57]. Overall, these methods are proven to improve PP glycemic control and reduce glucose variability, albeit with a non-significant reduction in HbA1c levels [57].

As in healthy subjects, also in T1DM patients the protein and fat content of the meal affects PP glucose profile. It has been demonstrated that the addition of fat to a meal tends to reduce the initial (2–3 h) glucose response, delaying glucose peak while increasing glucose levels in the late PP phase because of a slower GE [58,59]. In T1DM patients using insulin pump, more insulin was required following a high fat/ high protein meal, with some variation among patients; in addition, an extended, split insulin bolus best achieved PP glucose control [60]. In the context of a closed loop system (CLS), Wolpert et al. found that the addition of 50 g of fat increased insulin requirement by twofold [61]. More recently, Gingras et al. showed that CLS was able to mitigate late PP hyperglycemia following high fat/high protein meal through a 39% increase in basal insulin infusion in the 5-h post-meal period [62]. Of course, the complexity of these strategies and their impact on a patient’s everyday life should not be overlooked.

It is also interesting to note that the quality of fat influences the glucose response to a high GI meal, with high monounsaturated fat achieving lower glucose levels than saturated fat, as shown in a randomized cross-over study in T1DM patients on insulin pump [63]. The addition of protein to CHO also tends to delay PP glucose peak in T1DM patients [64,65]. Of note, the effect seems to be different when proteins are consumed alone or in addition to CHO; in the absence of CHO, a protein load as high as 75–100 g produces an increase in blood glucose [65]. In summary, in patients with T1DM, the macronutrient composition of the meal greatly influences not only the extent but also the “shape” of the PP glucose profile. Based on these findings, considering the overall macronutrient meal content into the calculation of pre-meal insulin bolus represents a challenge for the new automated insulin delivery systems and is an area of active investigation.

### 3.2. Pre-Meal Insulin

Type, dose, timing and method of insulin administration are important variables affecting PP glucose response. In healthy subjects, PP glycemia rises and reaches the peak at 60 min, hardly exceeds 140 mg/dL, and usually returns to basal values within 2–3 h. Simultaneously, there is a rapid increase in insulin concentration and action, within the first 30–120 min. This synchronized glucose-insulin response is lost in people with T1DM [66]. Thus, a major goal of insulin therapy is to rely on a premeal insulin that better mimics the kinetic of endogenous prandial insulin profile to effectively control PP glucose excursions. Rapid acting insulin analogues (RAIAs), i.e., lispro, aspart and glulisine—when used in basal-bolus regimens—resulted to be superior to regular human insulin in achieving this goal, since they restore insulin levels in a more physiological way [67,68]. Indeed, RAIAs induces a faster rise in plasma concentration, a higher peak concentration and a shorter subcutaneous residence time, which results in reduced PP glucose excursions and a lower risk of late hypoglycemia compared to regular human insulin [69,70]. However, the pharmacokinetic profile of RAIAs is still far from ideal, since the duration of subcutaneous absorption does not reproduce the prompt physiological increase in circulating insulin. This limit has been partially overcome by the recently developed fast-acting insulin aspart [67,68,71], a new insulin formulation characterized by faster subcutaneous absorption, faster onset of appearance into the bloodstream, and a 75% greater early glucose lowering effect than conventional aspart analogue [67,71,72]. The ONSET 1 registration trial and other clinical studies have shown the superiority of faster aspart in reducing PP glucose excursions (21 mg/dL at 1 h and 12 mg/dL at 2 h) and HbA1c levels compared to insulin aspart [73]. Despite the improvements achieved with the new insulin formulations, multiple daily injection therapy does not allow variations in the speed or duration of prandial insulin delivery [74]. Greater flexibility is guaranteed by the insulin pump therapy, which enables to modify prandial insulin delivery according to the composition of the meal so as to mitigate PP glucose excursions [75]. Several studies have evaluated PP glucose response following meals with different composition and different types of insulin boluses (single/quick bolus, two/split bolus, square wave bolus, dual wave bolus) [76,77,78,79]. Among the different types of boluses, the dual wave bolus is to be preferred for low GI meals and/or meals with a high fat content. In fact, the 3-h area under the curve (AUC) is lower with the use of double-wave bolus compared to the other types of boluses since it provides a better coverage of prolonged hyperglycemia. In contrast, high GI meals are associated with significant and prolonged upward PP glucose levels, regardless of premeal bolus type [74,77]. Although insulin pump therapy represents a remarkable advance in diabetes treatment, patients still struggle to achieve optimal overall and PP glycemic control, with a substantial number of patients presenting HbA1c > 8.0% [80]. A further step has been taken with closed-loop automated insulin delivery systems (CLS), in which insulin infusion is regulated by an algorithm based on CGM systems. Compared with conventional insulin pump therapy, CLS have been shown to improve overall glucose control although PP glucose excursions still remain a challenge [81,82,83], suggesting that other factors contribute to PP glucose profile.

## 4. Clinical Impact of Delayed Gastric Emptying and Therapeutic Implications

Achieving an optimal PP glucose control in T1DM patients remains a clinical challenge, despite the availability of more advanced insulin delivery systems and innovations in glucose sensing technologies having improved PP glucose control. In optimizing PP glucose control, it is worth measuring GE time since the rate at which carbohydrates are delivered to and absorbed by the small intestine affects the PP blood glucose profile. The finding that GE is delayed in a relevant proportion of young asymptomatic T1DM patients and that it impacts the magnitude and the “shape” of PP glycemic response has important implications for the management of pre-prandial insulin; in fact, a delayed GE can result in a mismatch between PP glucose elevation and the timing of insulin action, with consequent high variability in PP glucose profile and, possibly, increased risk of hypoglycemia. Along this line, we propose that GE assessment be included among the screening tests of diabetic complications. The availability of a non-invasive, validated, reproducible and inexpensive method, such as the ^13^C-octanoate breath test, would make the assessment of GE feasible and affordable in clinical practice [36,84]. Measurement of GE would be helpful not only to detect early manifestations of chronic diabetic complications but also guide the choice of pre-meal insulin therapy, thus making a step forward toward “tailoring” premeal insulin administration to patients’ characteristics. In patients with delayed GE who practice multiple daily injections, some options could be considered: (1) using regular insulin; (2) administering ultra-short insulin 20–30 min after meal ingestion; (3) splitting the insulin dose into two boluses. In all cases, the use of CGM would be fundamental to assess and compare the efficacy of the various insulin schemes. In patients on insulin pump, GE time could help identify the most suitable type of bolus (single bolus, double wave or square wave). A further step could be to include GE time into predictive algorithms for insulin delivery of the new closed-loop systems. In this context, the longer time to glucose peak associated with delayed GE could be handled through GE-guided temporary increase in basal insulin infusion. Ad hoc clinical studies are necessary to test the feasibility and efficacy of this proposal. The current literature on GE in relation to PP glucose regulation in T1DM is quite heterogeneous and too limited to allow a systematic analysis to be performed. Our review, although burdened by the limitation of a nonsystematic approach, has the merit of highlighting the important role of GE in PP hyperglycemia and providing further insight into the understanding and management of PP glucose control in T1DM patients. Studies quantifying the GE rate in relation to meal composition and different insulin delivery systems would provide valuable information to help achieve this goal.

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
