# Peer review of "Postprandial Glucose Control in Type 1 Diabetes: Importance of the Gastric Emptying Rate"

_nutrients, 2019, doi:10.3390/nu11071559_

Round 1
Reviewer 1 Report
The authors have revised this manuscript a few times based on previous round of comments from reviewers. The manuscript, in its current form, is a qualitative, non-formal review of PP glucose excursions in T1DM and the potential role and utility of GE.
The manuscript is does a good job of reviewing the 3 major areas affecting PP glucose in T1Dm.
Two areas that potentially could improve the manuscript is 1) in your final paragraph, detailing how you think the incorpation of GE into algorithms will change these algorithms (will it increase dosing rec's by 25%?) or at least, the types of studies that should be performed to answer these questions and 2) what are some limitations to your current reviews and opinions. Given that that is not a systematic review it is subject to potential bias and this should be considered.
Author Response
Reviewer 1
The authors have revised this manuscript a few times based on previous round of comments from reviewers. The manuscript, in its current form, is a qualitative, non-formal review of PP glucose excursions in T1DM and the potential role and utility of GE. The manuscript is does a good job of reviewing the 3 major areas affecting PP glucose in T1Dm.
Two areas that potentially could improve the manuscript is
1) in your final paragraph, detailing how you think the incorpation of GE into algorithms will change these algorithms (will it increase dosing rec's by 25%?) or at least, the types of studies that should be performed to answer these questions
Since PP glucose pattern consequent to delayed GE is characterized by a slower time to peak (resembling in some ways the glucose profile observed after a high fat/high protein meal), in the context of closed loop system a reasonable option could be to increase basal insulin infusion for some hours after the meal. We have added the following sentence: “In this context, the longer time to glucose peak associated with delayed GE could be handled through GE-guided temporary increase in basal insulin infusion” (lines 244-246).
2) what are some limitations to your current reviews and opinions. Given that that is not a systematic review it is subject to potential bias and this should be considered.
We added a sentence dealing with the limitation of the present review:” The current literature on GE in relation to PP glucose regulation in T1DM is quite heterogeneous and too limited to allow a systematic analysis to be performed. Our review, although burdened by the limitation of a nonsystematic approach, has the merit to highlight the important role of GE in PP hyperglycemia and provide further insight into understanding and management of PP glucose control in T1DM patients” (Lines 247-251).
Reviewer 2 Report
The manuscript has been improved substantially. The English requires minor attention.
Author Response
Reviewer 2
The manuscript has been improved substantially. The English requires minor attention.
The manuscript has been subjected to linguistic revision by a native English speaker.
This manuscript is a resubmission of an earlier submission. The following is a list of the peer review reports and author responses from that submission.
Round 1
Reviewer 1 Report
This review addresses an area of importance, but is poorly focussed and frequently repetitious, including information in other recent reviews, particularly a commentary (J Clin Endocrinol Metab 2018: 103; 3503-6) which is not referred to. – If the focus is on gastric emptying it should certainly be addressed before line 160! Currently postprandial glycaemic excursions have been shown to be of major relevance to HbA1c and their contribution is dependent on the baseline HbA1c level and, hence, primarily of importance to microvascular complications of diabetes – an independent contribution of PPG to microvascular disease is certainly possible, but remains to be clearly established. The major point is that the authors should focus on what is novel and minimise information that is established. The potential incorporation of gastric emptying in insulin algorithms is of particular interest - how will gastric emptying be measured?, will gastric emptying have advantages over CGMS?, what studies are required?, could prokinetics therapy be used to optimise the coordination between carbohydrate absorption and insulin delivery? As such, the review should be revised extensively.
The following are also relevant:
In a number of places, the authors could be accused, appropriately, of plagiarism eg line 22 – 28‘. ‘This is not surprising given that most individuals spend the majority of the day in the postprandial state and that …. before breakfast’.
Gastric emptying in adolescents with T1D may be accelerated (eg J Clin Endocrinol Metab 2015: 100; 2248 – 53).
A figure may be helpful.
Author Response
In light of the reviewer’s comments we have made the following changes:
The potential incorporation of gastric emptying in insulin algorithms is of particular interest - how will gastric emptying be measured ?
We have added a sentence on the method of GE measurement (lines 219-221). In addition, a paragraph on the therapeutic implications of GE assessment, and how GE time could be effectively used for the management of insulin therapy has been added (lines 217-230).
Will gastric emptying have advantages over CGMS?, what studies are required?,
In the revised version, we underline the fact that, for the impact of GE on PP glucose response to be fully appreciated, it is necessary to monitor blood glucose. Similarly, to evaluate the effectiveness of the various pre-meal insulin boluses, the use of CGM is essential (lines 226-227).
Could prokinetics therapy be used to optimise the coordination between carbohydrate absorption and insulin delivery? As such, the review should be revised extensively.
We have added a sentence stating that the use of prokinetic drugs has been tested in patients with gastroparesis with beneficial effects on PP hyperglycemia. To date, there is no data in patients with mildly prolonged GE time (lines 211-216).
The following are also relevant:
In a number of places, the authors could be accused, appropriately, of plagiarism eg line 22 – 28‘. ‘This is not surprising given that most individuals spend the majority of the day in the postprandial state and that …. before breakfast’.
The sentence has been removed.
Gastric emptying in adolescents with T1D may be accelerated (eg J Clin Endocrinol Metab 2015: 100; 2248 – 53).
The paper by Perano et al has been cited (ref. n. 67).
A figure may be helpful.
We have included a figure (Fig.1) illustrating the impact of GE and meal composition on postprandial glucose patterns.
Reviewer 2 Report
Overall comments:
This manuscript aimed to summarize the main factors contributing to PP glucose response and discuss difficulties in achieving optimal PP glucose control in type 1 diabetes. Overall, this manuscript contains some interesting discussion regarding the role of gastric emptying rate in PP glucose control. However, in general, the manuscript is a little hard to follow. In the abstract, the link between each section and GE is very clear; yet, in the remaining manuscript, it is sometimes unclear whether the authors are talking about PP glucose control in healthy subjects or in patients with T1D, and what is the link between each section.
Also, the introduction summarizes what will be discussed in the review, but the information is incomplete and thus a little hard to follow. It seems like the Introduction could be merged with the second paragraph on postprandial glucose control and clinical outcomes and focus on why postprandial glucose control is so important in type 1 diabetes, how it relates to overall glucose control, and what complications and clinical outcomes it is associated with.
In addition, the authors aimed to focus on gastric emptying, which is more novel, and this paragraph could have been expanded, while other sections could probably have been summarized.
Finally, throughout the manuscript, the impact of the macronutrient content (carbohydrates, proteins and lipids) and glycemic index should be defined separately, and more clearly, and the link between macronutrients and glycemic index with GE could be clarified.
Additional specific comments:
Abstract:
Line 11: In healthy subjects?
Introduction:
Lines 25-26: Unclear; overall glucose is not assessed by a HbA1c value <7%, it is assessed with HbA1c, with a target of <7%.
Lines 27-28: It seems strange to say that individuals spend the majority of the day in the postprandial state. What do authors mean by PP period? It is typically defined as 2-3 hours post-meal, although it can extend to 5-6 hours and it is thus true that some individuals spend a large amount, or a great proportion of time in the PP state, but the majority of the time sounds a little excessive.
Line 31: The energy content? Energy per se is not associated with PP glucose.
Line 34: Ref 4 - There are more suitable references specifically in T1D.
Line 43: strictly associated: unclear; the condition was dependent on other chronic conditions? Or it was only evaluated in patients with other conditions?
Postprandial glucose control and clinical outcomes:
Lines 55-57: Also specific to diabetes?
Lines 60-70: This paragraph is unclear. Why did the authors choose to cite these specific studies which are not in type 1 diabetes, and what is the link with the remaining manuscript?
Lines 70-71: There are novel studies on adjuvant drug therapy (e.g. GLP-1 agonists and SGLT-2 inhibitors; e.g. doi.org/10.1111/dom.12998, doi:10.2147/IJGM.S51665) and PP glucose control in diabetes and this possible therapeutic avenue has not been discussed in the manuscript.
Line 72: PP glucose control is not a novel parameter of glucose control.
Factors affecting postprandial glucose control:
Line 88: remove “the presence of antinutrients”, and why isn’t glycemic load or index included?
Line 90: Not only CHO-rich foods are classified. All foods have a GI. And would remove “as known”.
Line 93: Or a high content of protein or fat.
Line 96: Nutrient or macronutrient?
Lines 98-100: The authors should specify “in healthy subjects” since authors refer to insulin secretion. Yet, studies also showed that in T1D, protein and fat reduce the PP glycemic excursion. Update the references and rephrase.
Lines 166-167: In healthy subjects?
Lines 193-196: Any hypothesis why? Could GE be restored/improved?
Conclusions:
What is the day-to-day variability of GE and how do authors believe it could be included in closed-loop delivery, or in meal insulin calculations for patients using multiple injections or insulin pumps? Don’t the authors believe that the impact of GE is somewhat considered when insulin-to-carbohydrate ratios and basal doses are determined based on glucose profiles? What about adjuvant therapy for GE or for glucose control (e.g. GLP-1 agonists and SGLT-2 inhibitors)?
Author Response
We thank the Reviewer for his/her useful suggestions. On this basis, we have thoroughly reorganized the Introduction. In particular, we have moved the paragraph on the importance of PP glucose control from the section on “PP glucose and clinical outcomes” (that has been deleted) and shortened the section dealing with the factors involved in PP glucose regulation.
Finally, throughout the manuscript, the impact of the macronutrient content (carbohydrates, proteins and lipids) and glycemic index should be defined separately, and more clearly, and the link between macronutrients and glycemic index with GE could be clarified.
We have discussed the impact of macronutrients on PP glucose, separately in healthy and in T1DM subjects.
Additional specific comments:
Abstract:
Line 11: In healthy subjects?
We added healthy subjects
Introduction:
Lines 25-26: Unclear; overall glucose is not assessed by a HbA1c value <7%, it is assessed with HbA1c, with a target of <7%.
corrected
Lines 27-28: It seems strange to say that individuals spend the majority of the day in the postprandial state. What do authors mean by PP period? It is typically defined as 2-3 hours post-meal, although it can extend to 5-6 hours and it is thus true that some individuals spend a large amount, or a great proportion of time in the PP state, but the majority of the time sounds a little excessive.
In the reorganization of the Introduction, we have decided to remove the sentence.
Line 31: The energy content? Energy per se is not associated with PP glucose.
Corrected
Line 34: Ref 4 - There are more suitable references specifically in T1D.
Specific studies in T1DM have been cited (ref. 27,28)
Line 43: strictly associated: unclear; the condition was dependent on other chronic conditions? Or it was only evaluated in patients with other conditions?
The sentence has been removed.
Postprandial glucose control and clinical outcomes:
Lines 55-57: Also specific to diabetes?
Most of the reported studies have been performed in vitro by exposing target cells to acute increments in blood glucose.
Lines 60-70: This paragraph is unclear. Why did the authors choose to cite these specific studies which are not in type 1 diabetes, and what is the link with the remaining manuscript?
We cited the studies in T2DM because they are the only intervention studies evaluating whether the correction of PP hyperglycemia resulted in a reduction of CV outcomes. No long-term intervention study has been conducted in T1DM. We have shortened the paragraph dealing with studies in T2DM and added some references demonstrating an improvement of early atherosclerosis manifestations following glucose normalization in T1DM (lines 41-47, and Ref. 11).
Lines 70-71: There are novel studies performed on adjuvant drug therapy (e.g. GLP-1 agonists and SGLT-2 inhibitors; e.g. doi.org/10.1111/dom.12998, doi:10.2147/IJGM.S51665) and PP glucose control in diabetes and this possible therapeutic avenue has not been discussed in the manuscript.
We are aware that GLP-1 agonists effectively reduce PP hyperglycemia; however, at present there is no demonstration that this is the only mechanism responsible for the reduction of CV outcomes in T2DM patients.
Line 72: PP glucose control is not a novel parameter of glucose control.
We removed the sentence
Factors affecting postprandial glucose control:
Line 88: remove “the presence of antinutrients”, and why isn’t glycemic load or index included?
We removed “the presence of antinutrients”. In addition, we added a sentence on the glycemic load (lines 69-71, lines 83-85; ref.24, ref 32)
Line 90: Not only CHO-rich foods are classified. All foods have a GI. And would remove “as known”.
The expression GI food has been used throughout the manuscript.
Line 93: Or a high content of protein or fat.
It is now specified that foods can have a low GI because of a high content of fat or protein (lines 72-73)
Line 96: Nutrient or macronutrient?
Macronutrient
Lines 98-100: The authors should specify “in healthy subjects” since authors refer to insulin secretion.
We added “In healthy subjects”
Yet, studies also showed that in T1D, protein and fat reduce the PP glycemic excursion. Update the references and rephrase.
The impact of protein and fat on PP glucose in T1DM patients is now commented (lines 85-88)
Lines 166-167: In healthy subjects?
Added
Lines 193-196: Any hypothesis why? Could GE be restored/improved?
We do not have a clear hypothesis to explain the occurrence of delayed GE in T1DM patients who do not have signs of chronic diabetic complications in other districts. One could speculate that it is the consequence of glucotoxicity on gastric motor system.
As stated above to Reviewer 1, there is no information on the reversibility and/or the impact of prokinetic drugs in patients with mildly prolonged GE time. In patients with gastroparesis, prokinetic drugs have led some benefits both on GE and PP glucose response.
Conclusions:
What is the day-to-day variability of GE and how do authors believe it could be included in closed-loop delivery, or in meal insulin calculations for patients using multiple injections or insulin pumps?
For clinical purpose, the most used method for GE measurement is the 13C-octanoic acid breath test. It has a high degree of intra-individual reproducibility (CV 10-15%). The high reproducibility is another strength of this method, which would make GE time a reliable information to individualize insulin administration.
Don’t the authors believe that the impact of GE is somewhat considered when insulin-to-carbohydrate ratios and basal doses are determined based on glucose profiles?
We agree that the insulin-to-carbohydrate ratio and, hence, premeal insulin dose are in some way influenced by the rate of gastric emptying. However, the point is not so much the amount of insulin to be administered, rather the "timing" of insulin action.
What about adjuvant therapy for GE or for glucose control (e.g. GLP-1 agonists and SGLT-2 inhibitors)?
There is evidence that GLP1 agonists are able to improve PP hyperglycemia in T1DM patients; however, this therapy cannot be applied to T1DM patients who already have a delayed GE. These patients could potentially benefit of prokinetic drugs as reported in Lines 211-216, but no evidence is available at present.
Reviewer 3 Report
-The manuscript appears to have revisions due to the highlighting. If not, please remove the highlighting
-Consider adding the overall goal of your communication article in the introduction. Is it brief overview of the causes of postprandial glucose excursions in T1DM?
-line 99-101. There could be a better connection between T1DM, PP glucose and GI/GL in calculation insulin dose. There is a lot of good information in this specific section with a concluding sentence about using GI/GL in insulin dose calculations without much background on the current approach to insulin dose calculations in T1DM
-113-115. You say relative to regular insulin twice, this sentence could be adjusted to increase readability.
-Similar to your figure 1a, it may be useful to have such a figure for your section on pre-meal insulin and the different acting insulins.
-additionally, your figure 1b being paired with figure 1a is out of place given that the gastric emptying section does not occur until later.
-the GE section is the most interesting and best written. Perhaps the communication article should concentrate on this section (where you describe your own work) while potentially shortening other sections. The communication is lengthy as it currently stands and I still don't see the overall point of such communication versus creating a full review article.
Round 2
Reviewer 1 Report
The manuscript is improved, but I believe that further revision would be helpful. Specifically:
The first paragraph must mention the primary purpose of the review ie the relevance of gastric emptying to glycaemic control in type 1 diabetes and why this purpose is appropriate. Much of the detail in this paragraph could be abbreviated or omitted.
P2 Nutrient sequence should be referred to eg Nesti et al, Front Endocrinol 2019:10;11.
The section on gastric emptying should be earlier – a major point is that it has not been quantified in the vast majority of studies relating to the effects of meals and rapid acting insulin, which can be discussed subsequently.
A final paragraph summarising all the clinical implications would be helpful.
Lines 158 – 159 references relating to the effects of acute changes in glycaemia should probably be added.
Reviewer 2 Report
I believe the manuscript has been improved and most of the concerns have been addressed. However, I don’t feel convinced, from this manuscript, that GE should be included in CLS algorithms. I think the authors have expanded the discussion on GE, which is the novel and interesting aspect of this manuscript, but it is not quite clear whether GE needs to be considered for the calculation of prandial doses of insulin or closed-loop delivery, or if delayed GE simply needs to be identified and treated, when possible. Especially, for CLS systems, the algorithms are typically better at handling slow changes compared to acute changes in glycemia, and we could possibly expect them to handle quite well the lower peak and extended glucose excursions related to delayed GE?
Page 2; Lines 74-76: Dietary fat and protein do not reduce the postprandial response, they lower the peak and extend the postprandial excursion.
Page 2; Lines 89-91: More recent studies have looked at high fat and high protein meals in patients with T1D using CSII and in the context of CLS. The first one showed that following a high fat high protein meal, 65% more insulin was required (range 17%–124%) with a 30%/70% split over 2.4 hours (Diabetes Care 2016 Sep; 39(9): 1631-1634) while the study using CLS showed that high fat high protein meals required 39% more basal insulin in the 5-hours post-meal period compared to a standard meal (Diabetes Obes Metab 2018 Nov; 20(11): 2695-2699). These studies also suggest variability in insulin requirements with this kind of meal.
Page 3, Lines 99-101: Authors mention that “considering the overall macronutrient meal content into the calculation of premeal insulin bolus represents a goal of the new automated insulin delivery systems” which is perhaps an overstatement. This inclusion has been proposed in a non CLS context, but its usefulness and applicability has not been demonstrated. For example, the food insulin index (which is also applied to protein and fat foods) for insulin bolus calculation has been shown to improve postprandial glycemia following breakfast, with no improvement in other meals and in HbA1c (Diabetes Technol Ther. 2016 Apr;18(4):218-25) while a recent study showed no difference in terms of postprandial glucose excursion between carbohydrate counting and the food insulin index for insulin bolus calculation for a high fat high protein meal (Diabet Med. 2018 Oct;35(10):1440-1447). Also, the treatment burden associated with increasing the complexity of meal insulin bolus calculation would need to be considered.
Figure 1: I don’t feel like this figure was needed. Also, there are no axis titles. For fat and protein, the effect on postprandial glucose excursion is not quite the same, and the addition of both together typically further modifies the excursion.